# Theoretical and Experimental Studies on the Signal Propagation in Soil for Wireless Underground Sensor Networks

**DOI:** 10.3390/s20092580

**Published:** 2020-05-01

**Authors:** Hongwei Huang, Jingkang Shi, Fei Wang, Dongming Zhang, Dongmei Zhang

**Affiliations:** 1Department of Geotechnical Engineering, Tongji University, Shanghai 200092, China; huanghw@tongji.edu.cn (H.H.); 1710704@tongji.edu.cn (J.S.); 09zhang@tongji.edu.cn (D.Z.); dmzhang@tongji.edu.cn (D.Z.); 2Shanghai Institute of Disaster Prevention and Relief, Tongji University, Shanghai 200092, China

**Keywords:** wireless underground sensor networks, theoretical study, experimental study, signal propagation model

## Abstract

Wireless Underground Sensor Networks (WUSNs), an important part of Internet of things (IoT), have many promising applications in various scenarios. Signal transmission in natural soil undergoes path loss due to absorption, radiation, reflection and scattering. The variability and dynamic of soil conditions and complexity of signal attenuation behavior make the accurate estimation of signal path loss challenging. Two existing propagation models for predicting path loss are reviewed and compared. Friis model does not consider the reflection loss and is only applicable in the far field region. The Fresnel model, only applicable in the near field region, has not considered the radiating loss and wavelength change loss. A new two stage model is proposed based on the field characteristics of antenna and considers four sources of path loss. The two stage model has a different coefficient *m* in the near field and far field regions. The far field distance of small size antenna is determined by three criteria: 2 *D*^2^/λ, 5 *D*, 1.6 *λ* in the proposed model. The proposed two stage model has a better agreement with the field experiment data compared to Friis and Fresnel models. The coefficient *m* is dependent on the soil types for the proposed model in near field region. It is observed from experiment data that the *m* value is in the range of 0~0.20 for sandy soils and 0.433~0.837 for clayey silt.

## 1. Introduction

Wireless Underground Sensor Networks (WUSNs) have received much research interest in recent years since its promising potential applications in various scenarios, such as border intrusion control, underground structure monitoring, smart farming, environment pollution monitoring [1,2,3,4,5]. WUSNs, being underground, play an important role in the Internet of things (IoT) [6,7]. But four aspects of challenges are faced by the real application of WUSNs, namely power conservation, topology design, antenna design and environmental extremes [1]. Different from the traditional Wireless Sensor Networks (WSNs), high attenuation of wireless signals in soil is the most challenging aspect [8]. Natural soil is a composite of soil particles, air and water, and is usually inhomogeneous and anisotropic. Transmitted in such soils, signals undergo significant path loss due to absorption, reflection, refraction and scattering [1]. The variability and dynamic of soil conditions and complexity of signal attenuation behavior make the accurate estimation of signal path loss challenging. The estimation of signal path loss is crucial for antenna design, reliable communication links and node deployment optimization [9,10,11].

There are mainly two kinds of propagation models for predicting the signal path loss in soil in the literature: the Friis model [12,13,14,15,16] and the Fresnel model [17]. The complex dielectric permittivity of soil is the basis of propagation models. Existing propagation models all consider the natural soil as homogenous and isotropic media. Hence, the soil dielectric mixing models are needed to characterize the equivalent dielectric properties [18]. Peplinski and CRIM (Complex Refractive Index Model) are both empirical models to estimate the soil dielectric properties based on the physical parameters such as soil particle composition, water content and bulk density [19,20]. Peplinski model has received more recognition in previous studies, as compared to CRIM [12,13,14,15,16,21,22]. The time domain reflectometry (TDR) technique was also used to directly measure the soil dielectric properties instead of empirical prediction [23]. The attenuation constant *α* and phase shifting constant *β* can be determined by the soil complex dielectric permittivity. The path loss *L_p_* is expressed as a function of signal transmission distance *d* with key parameters including *α* and *β*.

The Friis model is modified from the free space path loss given by the Friis equation [14,15]. The path loss in dB is a logarithmic function of *d*. The Friis model fully considers the radiating energy loss of the EM (Electromagnetic) waves. The path loss caused by soil absorption and wavelength change are also considered in the Friis model [14,15]. The Friis model has a clear foundation and is widely used in the modeling of the underground channel. Modifications have been made to the Friis model using the TRD technique or a better soil dielectric mixing model to predict complex dielectric permittivity [23,24]. Nevertheless, the modified Friis model is still a one stage model and only valid in the far field region of the antenna [25]. The boundary between the near field and far field regions has not been clearly identified [26,27]. This leads to chaos in the usage of the Friis model. In addition, the antenna of the WUSN node is generally encapsulated in a box. The complex signal propagation and reflection among the antenna, the box and soil around the sensor node is not considered in the Friis model.

The Fresnel model has received much less attention from researchers in comparison to the Friis model. The Fresnel-CRIM model was proposed and validated by laboratory results to evaluate the signal attenuation of underground communication [17]. It is a linear model and has considered the signal reflection loss at the soil–air interface. Nevertheless, the Fresnel model lacks a strong theoretical basis and has not been verified by field experiments [23]. Field trials and laboratory tests of signal attenuation in different types of soils were conducted to compare the Friis model and Fresnel model. The Friis model is claimed to have a better performance than the Fresnel model in three locations of field trials [23]. However, another laboratory test concluded that the Friis model has better prediction in soils with higher permittivity, while the Fresnel model has a better prediction in soils with lower permittivity [28].

Both existing propagation models have pros and cons from what has been discussed. This paper develops a new two stage model which aims to tackle the disadvantages of the existing models. The main contributions of this paper are simplified as follows: (1) prove that the attenuation factor α for the Friis model and the Fresnel model are equivalent for the same soil; (2) identify the applicable conditions for the Friis model and the Fresnel model, based on the field characteristics of the antenna; (3) develop a new two stage model considering four sources of path loss and incorporate the advantages of existing models into the new model; (4) clarify the far field distance with three criteria for node antenna; (5) verify the proposed model using field experiments covering different frequencies, different soil types, different water contents and different antenna field regions.

## 2. The Related Path Loss Model for EM Waves in Literatures

### 2.1. Friis Model (Modified Friis Equation)

The Friis model is modified from the Friis equation in free space when EM waves are transmitted in soils. Path loss quantifies the EM wave attenuation with the transmission distance. When the antenna gains are not included, then path loss in free space can be simplified to Equation (1) [14].
(1)$$L_{p}=-147.6+20\log_{10}(d)+20\log_{10}(f),$$
where *f* is the operation frequency of EM waves. The free space propagation model for Equation (1) is valid only when the distance *d* lies in the far field (Fraunhofer) region. The far field region is defined as the region beyond the far field distance *d_f_* which is determined by Equation (2) [25].
(2)$$d_{f}=2D^{2}/\lambda ,\;d_{f}>>D,\;d_{f}>>\lambda ,$$
where *D* is the largest physical linear dimension of antenna, *λ* is the wavelength in free space. EM waves transferring in soils induces additional path loss *L_m_* which consists of two parts *L_m1_* and *Lα*_0_ Then total path loss in dielectric materials is expressed in Equation (3) [14].
(3)$$L_{p}=6+20\log_{10}(d)+20\log_{10}(\beta_{0})+8.69\alpha_{0}d,$$
where *d* is the propagation distance in meters, and the attenuation constant *α*_0_ and phase shift constant *β*_0_ are determined by Equations (4) and (5), respectively.
(4)$$\alpha_{0}=2\pi f\sqrt{\frac{\mu \varepsilon_{0}K^{\prime }}{2}\left[\sqrt{1+{\left(\frac{K^{\prime \prime }}{K^{\prime }}\right)}^{2}}-1\right]},$$
(5)$$\beta_{0}=2\pi f\sqrt{\frac{\mu \varepsilon_{0}K^{\prime }}{2}\left[\sqrt{1+{\left(\frac{K^{\prime \prime }}{K^{\prime }}\right)}^{2}}+1\right]},$$
where *f* is the EM wave frequency, *μ* is the magnetic permeability of soil, *K*′ and *K*″ are the real and imaginary parts of relative soil complex dielectric constant *K** (*K** = *K*′ + *jK*″), respectively.

Note that the path loss model in soils given by Equation (3) is determined by EM wave frequency and soil electromagnetic constants (*μ*, *K**). What is more, the magnetic permeability *μ* for most soils which do not contain ferromagnetic substances is equal to free space permeability *μ*_0_, and can be assumed to be a constant for different types of soils.

### 2.2. Fresnel Model

The Fresnel model proposed by Bogena et al. is a semi-empirical model based on many years of practices of electromagnetic measurement techniques such as time domain reflectometry (TDR) and ground penetration radar (GPR) [18]. These methods provide a wide knowledge of soil electromagnetic properties and give an estimate of EM wave attenuation in soils. The soil attenuation constant *α_c_* is obtained according to transmission line theory giving in Equation (6) [29].
(6)$$\alpha_{c}=\frac{60\pi (2\pi f\varepsilon_{0}\varepsilon^{\prime \prime }+\sigma_{b})}{\sqrt{\frac{\varepsilon^{\prime }}{2}\left[\sqrt{1+{\left[(\varepsilon^{\prime \prime }+\frac{\sigma_{b}}{2\pi f\varepsilon_{0}})/\varepsilon^{\prime }\right]}^{2}}+1\right]}},$$
where *f* is the operation frequency, *ε*_0_ is the dielectric permittivity of free space, *σ_b_* is the bulk electric conductivity, and *ε*′ and *ε*″ are the real and imaginary parts of soil dielectric permittivity at frequency *f*, respectively. The total electromagnetic wave attenuation in soils *A_tot_* is added by soil attenuation and reflected energy loss *R_c_* in Equation (7) [17].
(7)$$A_{tot}=8.69\alpha_{c}d+R_{c},$$

The path loss *R_c_* due to signal reflection at the interface between soil and air can be calculated using the reflection coefficient as shown in Equation (8).
(8)$$R_{c}=10\log_{10}(\frac{1}{1-R^{2}}),$$
where *R* is reflection coefficient, and can be obtained using the soil equivalent dielectric permittivity as shown in Equation (9).
(9)$$R=\left|\frac{1-\sqrt{K^{*}}}{1+\sqrt{K^{*}}}\right|,$$
where *K** is the relative soil complex dielectric constant.

### 2.3. Comparison between the Two Models

Two path loss models for EM wave propagation in soil are presented above. Some observations can be drawn from the comparison of the two models: (1)The Friis model for estimating EM wave path loss contains three parts: linear part, logarithm part and constant part. The linear part “8.69*α*_0_*d*” stands for the wave attenuation due to soil absorption. The logarithm part “20log_10_(*d*)” stands for the attenuation due to wave free propagation. The Fresnel model is a totally linear path loss model which only contains the linear part and the constant part. The Fresnel model is based on transmission line theory while the Friis model is based on the free space propagation model.(2)The key parameters for the two models are the attenuation constants *α*_0_ and *α_c_*, respectively. Both attenuation constants are determined by the soil electromagnetic parameters. *α*_0_ is determined by soil complex dielectric constants *K** (*K*′, *K*″) while *α_c_* is determined by soil dielectric constants *ε* (*ε*′, *ε*″) and soil bulk conductivity *σ_b_*. The relationship between the soil complex dielectric constants *K** and soil dielectric constants *ε* and bulk conductivity is given by Equation (10) [30]. From this definition, the following relationship can be obtained: *K*′ = *ε*′, and *K*″ = *ε*″ + *σ_b_*/2*πfε*_0_.
(10)$$K^{*}=K^{\prime }+j(\varepsilon^{\prime \prime }+\sigma_{b}/2\pi f\varepsilon_{0}),$$(3)The attenuation constants *α*_0_ and *α_c_* can be proven to be equal. The details are as follows. Equation (6) can be expressed as Equation (11) considering the relationship shown in Equation (10).
(11)$$\alpha_{c}=2\pi f\frac{60\pi K^{\prime \prime }\varepsilon_{0}}{\sqrt{\frac{\varepsilon^{\prime }}{2}\left[\sqrt{1+{\left(K^{\prime \prime }/\varepsilon^{\prime }\right)}^{2}}+1\right]}},$$

Then assume that Equation (11) is equal to Equation (4), and we have,
(12)$$2\pi f\frac{60\pi K^{\prime \prime }\varepsilon_{0}}{\sqrt{\frac{\varepsilon^{\prime }}{2}\left[\sqrt{1+{\left(K^{\prime \prime }/\varepsilon^{\prime }\right)}^{2}}+1\right]}}=2\pi f\sqrt{\frac{\mu \varepsilon_{0}\varepsilon^{\prime }}{2}\left[\sqrt{1+{\left(K^{\prime \prime }/\varepsilon^{\prime }\right)}^{2}}-1\right]},$$
(13)$$60\pi K^{\prime \prime }\varepsilon_{0}=\frac{\varepsilon^{\prime }}{2}\frac{K^{\prime \prime }}{\varepsilon^{\prime }}\sqrt{\mu \varepsilon_{0}},$$
(14)$$\sqrt{\frac{\mu }{\varepsilon_{0}}}=120\pi ,$$

As *μ = μ*_0_ = 4*π* × 10^−7^ H/m, *ε*_0_ = 1/(36*π*) × 10^−9^ F/m, Equation (14) is naturally fulfilled. Hence, the attenuation constants *α*_0_ and *α_c_* is be proven to be equal. The uniform attenuation constant *α* is used in the rest of this paper.

## 3. Proposed Soil Propagation Model

Different to the long transmission distance in air, the wireless signal transmission in soil is limited to a few meters. In such a short transmission distance, the far field assumption for the Friis model is questionable. A reasonable wireless signal propagation model in soil is needed to consider the field characteristic of the antenna as shown in Figure 1. Based on the field characteristics, the field is classified into three regions; the Reactive near field region, the Radiating near field region (Fresnel region) and the Far field region (Fraunhofer region). The boundaries are defined by many studies, and many criteria are established to clarify different regions. Most commonly used and widely accepted criterion for the boundary between the Reactive near field and the Radiating near field is *λ*/2*π* [27]. The boundary between the Fresnel region and Fraunhofer region is 2*D*^2^/*λ*. Measurement of antenna parameters reveals that no strict transition exists between these regions.

Reactive near field region is the immediate region around the antenna where electric and magnetic fields are temporally out of phase and result in reactive energy. The energy is predominately stored in this field and does not propagate. The reactive near field is very small, only 0.11 m for the antenna operating at 433 MHz. Far field region is the region around antenna where electric and magnetic fields become in-phase temporally. The far field only radiates energy and no energy is stored in this region. Radiating near field region lies between the Reactive near field and far field. The energy is partially stored and partially radiating. The energy stored is less than that which is radiated in the Fresnel region.

Based on the field characteristics of the antenna, a more in-depth sight of the Friis and Fresnel models can be obtained. The Friis model is only valid when the transmission distance lies in the far field region. The far field distance is very vague, as given in Equation (2). The blind use of the equation of *d_f_* = 2*D*^2^/*λ* is not fulfilling. In other words, Friis model cannot be correctly used if the far field distance is not clearly determined. Unlike the communication in air, the far field distance will exert a big effect on the accurate estimation of signal path loss in underground communication. The Fresnel model is more suitable to be used in the Fresnel region of antenna. However, the existing Fresnel model does not consider the radiating energy loss and the wavelength change loss in the Fresnel region. The Fresnel model is obviously not reasonable in the far field region. The attenuation constants for the two models are proven equal, which reveals no conflict between the two models.

A new path loss model has to be established since the existing models have disadvantages. According to the field characteristic of antenna, the wireless signal propagation model is a two stage model; near field model and far field model. The Reactive near field is omitted in the path loss model due to no radiating energy in this field and is very limited in range around the antenna. The new model considers the path loss caused by four sources as shown in Equation (15).
(15)$$L_{p}=L_{d}+L_{r}+L_{w}+L_{Rc}$$
where *L_d_* is the path loss induced by soil absorption, *L_r_* is the path loss induced by radiating, *L_w_* is the path loss induced by wavelength change from air to soil, *L_Rc_* is the path loss induced by reflection at the soil–air interface. The two stage model is expressed in Equation (16) marked by a far field distance *d_f_*.
(16)$$\begin{array}{ll} L_{p} & =8.69\alpha d+m\cdot 20\log_{10}(d)+20\log_{10}(\beta )+6+R_{c}\\ & =8.69\alpha d+20\log_{10}(\frac{d^{m}\beta }{\sqrt{1-R^{2}}})+6,\\ & 0\leq m<1,\;d\leq d_{f};m=1,d>d_{f} \end{array},$$
where *α* is the soil attenuation constant, *d* is the signal propagation distance, *β* is the phase shifting constant, *R* is the reflection coefficient obtained by Equation (9), *d_f_* is the far field distance. The coefficient *m* has a different value in the near field and far field regions. 0 ≤ *m* <1 stands for the two stage model in the near field region and *m* = 1 stands for the two stage model in the far field region. Coefficient m considers the partially radiating loss in the Fresnel region and is usually seen as a fitting parameter.

The two stage model has a key issue in the determination of far field distance. The commonly used criteria 2*D*^2^/*λ* is only suitable for large size antenna. However, 433 MHz is typically used for WUSNs to decrease the attenuation. The antenna size is 0.17 m for a wireless sensor node operating at 433 MHz. So the far field distance is clarified in the new model considering the small size antenna of the sensor node. The far field distance is determined by three criteria as shown in Equations (1)~(19) [26].
(17)$$d_{f}=2D^{2}/\lambda ,$$
(18)$$d_{f}>5D,$$
(19)$$d_{f}>1.6\lambda ,$$

The three criteria as a function of electrical size of antenna are shown in Figure 2. From the figure, it can be seen that the widely used formula *d_f_* = 2 *D*^2^/*λ* is only valid when the antenna size *D* is larger than 2.5 *λ*. If the antenna size D < 0.32 *λ*, then the far distance *d_f_* = 1.6 *λ.* If the antenna size is in the range of 0.32 *λ* < *D* < 2.5 *λ*, then the far field distance *d_f_* = 5 *D*. The antenna size is 0.17 m for a wireless sensor node operating at 433 MHz. The wavelength in the free space is 0.693 m, D/*λ* = 0.25 < 0.32. So the far field distance is 1.6 *λ* = 1.11 m. The wavelength will be decreased when the signal transmitting from free space into soil. The wavelength in soil is calculated using the formula *λ* = 2π/*β*_0_. *β*_0_ is the phase shifting constant given by Equation (5). For most soils, the calculation results show that the antenna size lies in the range of 0.32 *λ*< *D* < 2.5 *λ*, the far field distance *d_f_* = 5 *D* = 0.85 m. 

The proposed two stage model is compared with the existing models in a qualitative way, as shown in Figure 3. Two key contributions of the proposed model can be concluded. Contribution 1: the proposed model is a two stage path loss model which is applicable both in the near field region and far field region. The Friis model is only applicable in the far field region and the Fresnel model is only applicable in the near field region. The far field distance is redefined in the proposed model considering the small antenna size of the sensor node. Contribution 2: the proposed model consider four sources which result in signal path loss. The loss caused by soil absorption is all considered in different models. However, the Friis model does not account for the reflection loss at the soil–air interface. The Fresnel model does not consider the wavelength change loss and the radiating loss.

The proposed two stage model is compared with the existing models in a quantitative way as shown in Figure 4. The signal frequency is set at 433 MHz and the soil equivalent complex permittivity is 13.25–2.18 *j* in the simulation. Coefficient *m* is set at 0.5 in the near field region for the proposed model. It can be seen that the Friis model has a larger estimation of path loss than the proposed model as the full consideration of radiating loss in the near field. The Fresnel model has a smaller estimation of path loss to that of the proposed model, as the radiating loss and wavelength change loss are not considered in the Fresnel model.

## 4. Onsite Experiment Setup

Onsite experiments of EM wave propagation in soils were conducted to evaluate the proposed model. This paper set up the experiment along the vertical direction as shown in Figure 5. The system was composed of two wireless sensor nodes, one gateway and one laptop. Node 1 was buried at the bottom of the hole and Node 2 was placed at the surface of the backfilled soil. The distance between Node 1 and Node 2 was changed as the covering soil increased. The specifications for wireless sensor nodes are listed in Table 1.

A gateway was used as a key unit in the Wireless Sensor Network system. It was responsible for the command issuing (such as T, F modifications) to, and data collection from, all of the nodes involved in a mesh network; meanwhile, it forwarded the data and system information to the remote server via a mobile network. The sensor node contained the same radio module as in the gateway with the mesh protocol embedded. In addition, it contained industrial typed D Cell batteries as the power source, the power management module, an MCU (Microcontroller Unit) chip and MEMS (Micro-Electro-Mechanical System) sensor chips, such as temperature and tilt. Dipole antennas were applied and maintained as vertically polarized throughout the whole experiment. The antenna orientations of Node 1 and Node 2 were the same. At each time interval, Sensor nodes formed a mesh topology and sent data to the gateway. The wireless protocol (namely, WISENMESHNET^®^) used in this experiment was designed specifically for a Wireless Sensor Network monitoring system. It was based on IEEE802.15.4. The sensor Node 2 received the signal sent by Node 1, recorded the RSSI (Received Signal Strength Index) and transferred this information to the gateway. The gateway transmitted the information to the remote server via the mobile network and the experiment data was obtained from the laptop. The steps for the onsite experiment are shown in Figure 6. The hole diameter is 0.5 m to reduce the signal reflection at the hole wall as much as possible.

Four groups of experiments with three different soil types were conducted to obtain as much experiment data as possible. The physical and electrical parameters of the soils backfilled into the hole are given in Table 2. The physical soil properties were measured by laboratory soil tests and the bulk electric conductivity was measured using the TDR technique. The dry sand had no water content because the sand has been dried in the sun and this result had been proven by the oven drying method. For wet sand, it was a challenge to make sure there was uniform distribution of water because the sand had a very good conductivity of water. In this experiment, some measures were taken to make the distribution of water as uniform as possible. The VWC (Volumetric water content) of wet sand in this paper was 4.9%, which was still a very low water content. The wet sand was backfilled in the hole, layer by layer as quickly as possible, and the experiment was rapidly completed in two to three hours to reduce the dispersion of water. The experiment temperature was maintained at approximately 20 °C.

Measurements of the RSSI were repeated three times to investigate repeatability and reliability. The RSSI was measured by the internal radio chip, and the accuracy was +/− 1 dBm. The RSSI value for each measurement was confirmed to be stable according to the criteria that the maximum difference of the three RSSI values should be no larger than 1 dBm. The RSSI value used in the analysis was the last one of the three. During the experiments, the wireless system was set at 1 min time interval, i.e., one data packet was generated and sent to the parent nodes/gateway at every minute. The specific number of packets was determined by the experiment time needed for every group. It took approximately two to three hours to complete every experiment group.

## 5. Experiment Results and Discussion

The experiment results mainly include the node distance (between Node 1 and Node 2) and RSSI for Node 2 as shown in Figure 7 and Figure 8. The acquired experiment data is given in Table A1 in the Appendix A. The attenuation constant *α* is calculated using the Peplinski model according to the soil physical parameters given in Table 2. Peplinski model is a widely accepted semi-empirical model for predicting the dielectric properties using soil physical parameters [27,28]. In Figure 7, the predicted attenuation constant *α* is 17.42 for clayey silt. The Friis model and the proposed model are compared in the figure. The proposed model has a better fitness to the experiment data for clayey silt compared to Fresnel model and Friis model. Due to the large attenuation and limited transmission distance for clayey silt, the linear part of “8.69*αd*” predominates in the total path loss. So the predicted path loss of different models does not have a very large deviation. The transmission distance is less than 0.85 m and the proposed model is in the near field region. The coefficient *m* is 0.837 for clayey silt group (a) and 0.434 for group (b). The different coefficient *m* is mainly caused by the experiment condition such as the possible nonuniformity of backfilled soil and the subtle change of antenna orientation.

The two stage proposed model behaves well for the wet sand as shown in Figure 8a. The far field distance is 0.85 m calculated by the proposed model. The path loss increases at a more rapid rate when the propagation distance exceeds the boundary between near field and far field region. Neither the Fresnel model nor the Friis model alone can explain such a phenomenon. The two stage proposed model is essential for interpreting the path loss observed in the field experiment. The calculated attenuation constant *α* for dry sand is equal to zero. This can be verified by the experiment data as shown in Figure 8b. The Friis model cannot explain the phenomenon of no path loss in dry sand. The Fresnel model cannot answer the question: “Now that the attenuation constant equals to zero, does it mean that the propagation distance is infinite in dry sand?” The two stage model shows that the energy should radiate in the far field region. In addition, the Fresnel model does not consider the wavelength change loss. This illustrates a lower path loss prediction compared to observed data in Figure 8b.

Coefficient of Determination (*R*^2^) and Root Mean Squared Error (RMSE) are employed to compare the goodness of fit for these theoretical models. For a set of experiment data (*y*_1_, *y*_2_, …, *y_i_*, …, *y_n_*), a function *f* is used to fit the data and a set of fitted data (*f*_1_, *f*_2_, *…f_i_*, …, *f_n_*) is then obtained. The mean of the experiment data, the Sum of the Squares of Residuals (SSres) and the Total Sum of Squares (SStot) are defined using Equations (20)–(22).
(20)$$\bar{y}=\frac{1}{n}\sum_{i=1}^{n}y_{i},$$
(21)$$SSres=\sum_{i=1}^{n}{\left(y_{i}-f_{i}\right)}^{2},$$
(22)$$SStot=\sum_{i=1}^{n}{\left(y_{i}-\bar{y}\right)}^{2},$$

Then, the *R*^2^ and RMSE can be defined as follows:(23)$$R^{2}=1-\frac{SSres}{SStot},$$
(24)$$\mathrm{RMSE}=\sqrt{\frac{1}{n}\sum_{i=1}^{n}{\left(y_{i}-f_{i}\right)}^{2}}=\sqrt{\frac{1}{n}SSres},$$

The values of *R*^2^ and RMSE for each experiment group are listed in Table 3. Both indicators support that the proposed two stage model has a better performance than the Friis and Fresnel models. The very small value of SStot results in *R*^2^ = 0.01 for dry sand, but the RSME is still very small. The coefficient *m* in the proposed model is found to be variated for each experiment group. It seems that the *m* value equals 0 for sand, but this needs more experimentation for verification. The experiment condition is likely to have an effect on the *m* value as shown by the two clayey silt groups.

As only four groups of experiments have been conducted, more field experiments from the literature are investigated in this paper [23,31,32]. Figure 9 shows the experiment scheme from the literature [23]. The experiment was conducted in the horizontal direction at three separate locations. Wireless sensor nodes operating at 433 MHz were buried at a depth of 500~600 mm to avoid the signal reflection at ground surface. Soil dielectric properties of different sites were measured by TDR technique, as shown in Table 4.

The experiment data is plotted in Figure 10. Experiment data for location A is omitted as there are only two data points. The *R*^2^ is 0.97 and RMSE is 1.74 for the proposed model in location B as shown in Table 5. The coefficient *m* is 0.50 for the near field region. The Friis model and the Fresnel model have a bad performance in predicting the path loss. For location C, the signal is mainly propagated in the far field region as the far field distance is 0.85 m. The *R*^2^ is 0.93 and RMSE is 2.93 for the proposed model in location C. The Friis model has a relatively good performance in this experiment because four data points are in the far field region. The *R*^2^ is 0.74 and RMSE is 5.50 for Friis model in location C. However, the improper determination of far field region leads to the path loss prediction error. The proposed two stage model shows a better agreement with the two groups of experiment data compared to the Friis and the Fresnel models.

The field test was also conducted at the frequency of 433 MHz in the literature [31]. The buried depth for sensor nodes was fixed at 0.4 m and the inter-node distance was variated from 0.1 m to 1 m. In the trial, the soil composition was 15% clay particle, 35% silt particle, 50% sand particle. The bulk density was 1.5 g/cm^3^, and the solid soil particle density was 2.6 g/cm^3^. Experiments realized in dry and wet conditions correspond to volumetric water content of 10% and 30%, respectively. Comparison of experiment results and the propagation models are given in Figure 11. The far field distance is 0.85 m for both experiments. The data points are in the near field and far field regions. In these conditions, the proposed two stage model has a much better prediction performance compared to the other two models. Both *R*^2^ of the two groups of data reached 0.95 for the proposed two stage model as shown in Table 5.

The above field experiments were all conducted at the frequency of 433 MHz. Another field experiment at the frequency of 2.4 GHz was employed to investigate the proposed two stage model at a different frequency [32]. Two types of sand from two locations were used as an underground medium. The electric properties of two types of sand were *σ* = 37.61 mS/m, *ε_r_* = 19 for 8% wet sand and *σ* = 123.61 mS/m, *ε_r_* = 30 for 15% wet sand. The wireless sensor nodes were buried in the depth of 1.40 m to avoid the possible EM interferences. The signal wavelength was 0.125 m at 2.4 GHz and the antenna dimension was *D* = 0.03 m. Far field distance was obtained using the criterion *d_f_* = 5D = 0.15 m. Consequently, the signal transmission path was clearly in the far field. The comparison of experiment data with the propagation models are shown in Figure 12. The Fresnel model had a bad performance in the far field region. The performance of the Friis model was still not as good as the proposed model because of the neglect of reflection loss *R_c_*. The *R*^2^ is 0.89 and RMSE is 2.79 for the proposed model in 8% wet sand, and the *R*^2^ is 0.98 and RMSE is 2.22 for the proposed model in 15% wet sand, as shown in Table 5. Thus, the proposed two stage model is also verified at 2.4 GHz.

The comparison between the experiment results from the literature and propagation models are summarized in Table 5. Ten groups of field experiments have been provided for the verification of path loss models. These ten groups of experiments cover different soil types, different water content, different operating frequency and different field regions of antenna. It can be concluded that the proposed two stage model has a better agreement with various field experiments than the Friis and Fresnel models. The coefficient *m* of the proposed model in the near field region is correlated with the soil types. For sandy soils, the *m* value is relatively small and in the range of 0~0.20. For clayey silt, the *m* value is much larger and in the range of 0.433~0.837.

## 6. Summary and Outlook

The wireless signal transmission in soil is a great challenge due to the extreme attenuation. The path loss prediction is very important in the design and deployment of WUSNs. The existing two theoretical propagation models are reviewed and compared in this paper. Both models have pros and cons. The field characteristics are thought to be a critical point to investigate the soil propagation in WUSNs. A realistic soil propagation model should be founded on the antenna field characteristics and consider the path loss induced by all possible sources. A new two stage model is proposed and verified by field experiment in this paper. Some important points are concluded as follows:(1)The field around the antenna is generally classified into three regions; Reactive near field region, Radiating near field region and far field region. The electromagnetic energy is restored in the Reactive near field region. The energy is thoroughly radiating in the far field region, and the energy is partially restored and partially radiating in the Radiating near field region. The Friis model is only valid in the far field region and the Fresnel model is only valid in the near field region. The attenuation constant α for the two models is proven to be equal in the same soil.(2)A two stage propagation model in soil is proposed in this paper. The two stage model covers the path loss induced by four sources; soil absorption, radiation, reflection and wavelength change. The boundary between two stages is the far field distance, which is determined by the maximum of the three criteria; 2 *D*^2^/λ, 5 *D*, 1.6 *λ*.(3)The proposed propagation model is verified by the field experiments covering different soil types, different water content, different operating frequency and different field regions of antenna. The *R*^2^ of the proposed model all exceeds 0.9 for ten groups of field experiment data and is 0.89 for the other one group. The proposed two stage model shows a better fitness compared to the existing Friis model and Fresnel model.(4)The coefficient *m* for the proposed model in near field region is dependent on the soil types. It is observed that *m* value is relatively small for sandy soils with the range of 0~0.20, and this value is much larger for clayey silt with the range of 0.433~0.837.

The coefficient m for the proposed model needs further study as it is considered as an undetermined factor in the current model. Coefficient *m* has a close relationship with the soil composition from the experiment data. This relationship has to be verified by more field experiments and expressed in a more quantitative way. All the mentioned path loss models have the same assumption that natural soils are considered as homogenous and isotropic. This assumption is challenged by the possible scattered rocks, plant roots and varying soil properties in space. These will lead to signal scattering and multipath fading, which are not considered in the proposed two stage model.

## Figures and Tables

**Figure 1 sensors-20-02580-f001:**
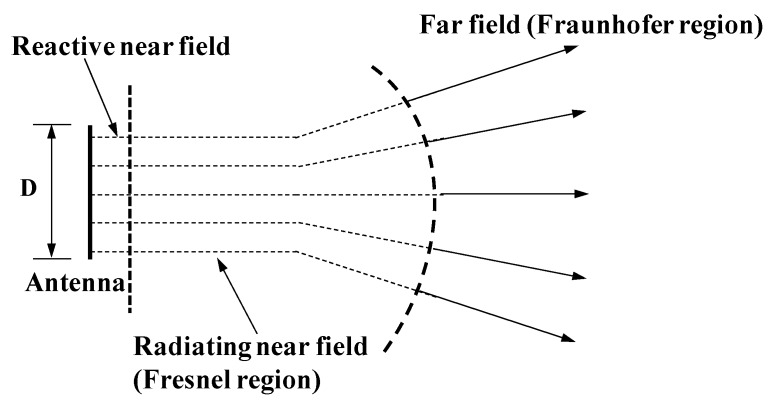
Field characteristic of antenna.

**Figure 2 sensors-20-02580-f002:**
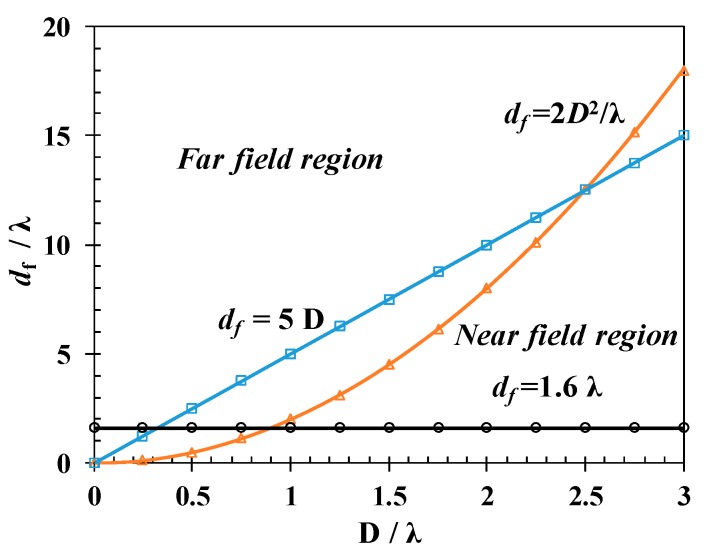
The three criteria of the far region as a function of normalized electrical antenna size (*D*/*λ*)**.**

**Figure 3 sensors-20-02580-f003:**
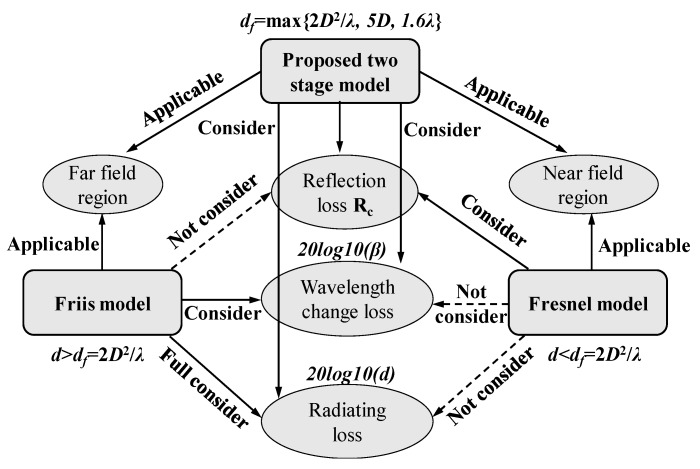
Comparison among the proposed model, the Friis model and the Fresnel model in a qualitative way.

**Figure 4 sensors-20-02580-f004:**
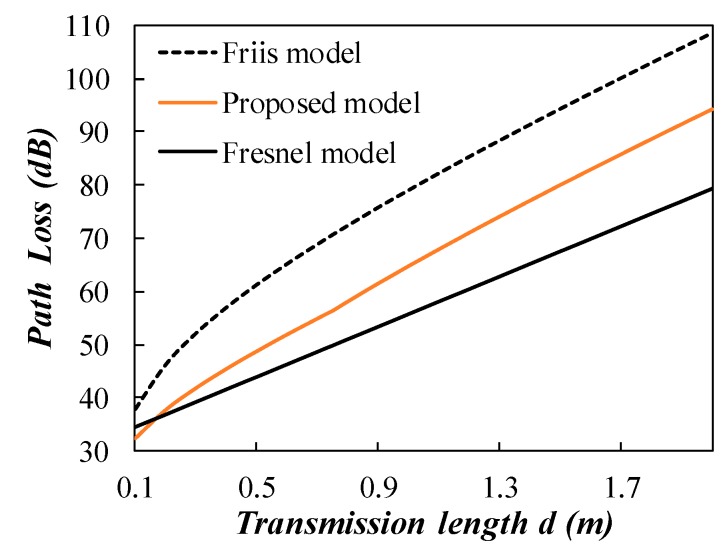
Comparison between the Friis model and the Fresnel model; *f* = 433 MHz, soil equivalent complex permittivity 13.25–2.18 *j*.

**Figure 5 sensors-20-02580-f005:**
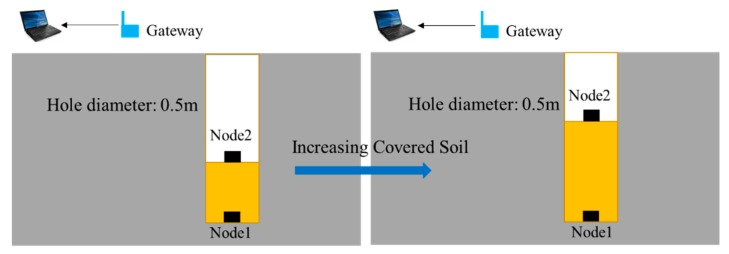
Schematic of the onsite experiment in this paper.

**Figure 6 sensors-20-02580-f006:**
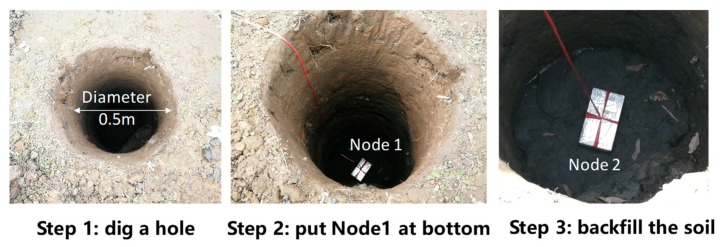
Steps for the onsite experiment.

**Figure 7 sensors-20-02580-f007:**
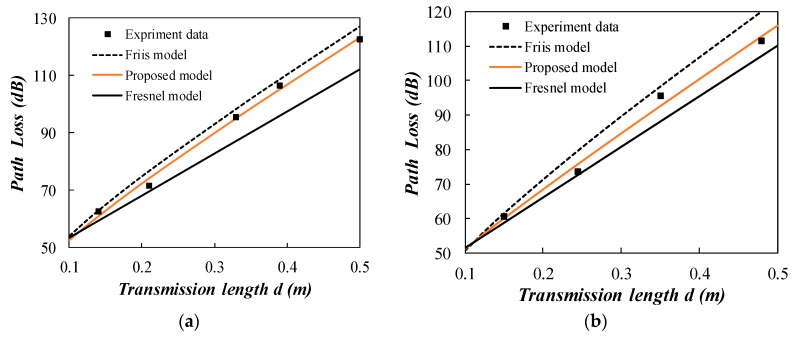
Comparison of the measured path loss with the propagation models (clayey silt). (**a**) group 1; (**b**) group 2.

**Figure 8 sensors-20-02580-f008:**
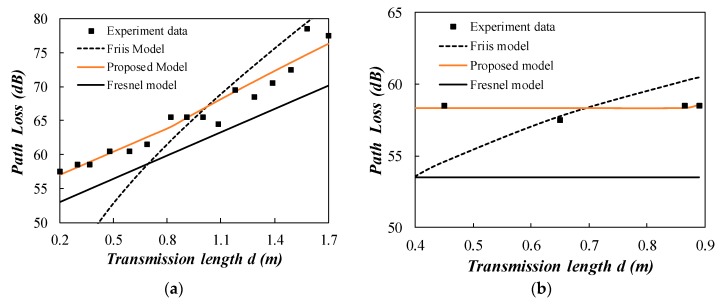
Comparison of the measured path loss with the propagation models (sand). (**a**) wet sand; (**b**) dry sand.

**Figure 9 sensors-20-02580-f009:**
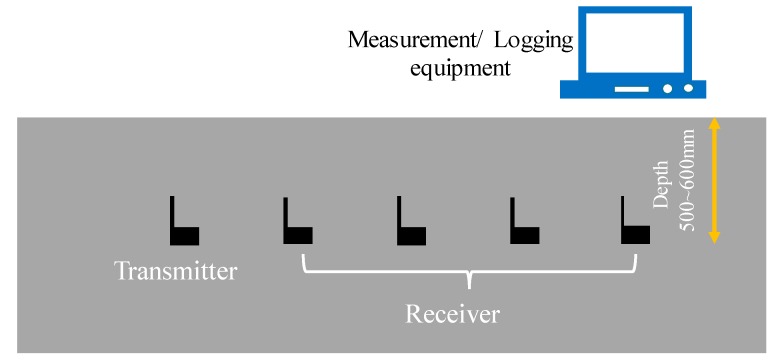
Schematic of the RF (Radio Frequency) trials from literature [23].

**Figure 10 sensors-20-02580-f010:**
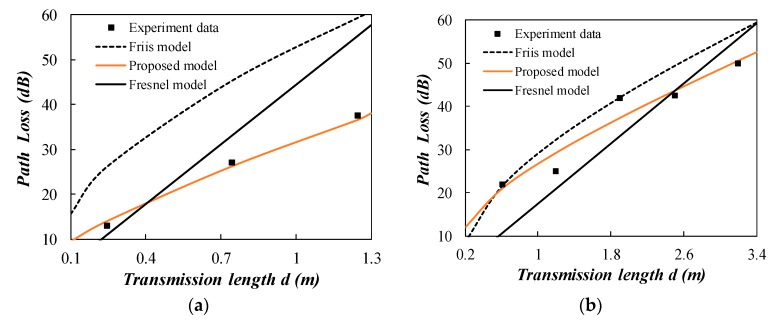
Comparison of the propagation models with experiment data from the literature [23]. (**a**) location B; (**b**) location C.

**Figure 11 sensors-20-02580-f011:**
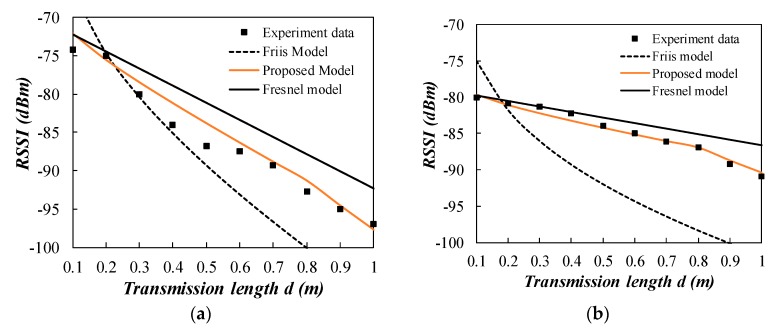
Comparison of the measured experiment data from the literature [31] with the propagation models. (**a**) VWC = 30%; (**b**) VWC = 10%.

**Figure 12 sensors-20-02580-f012:**
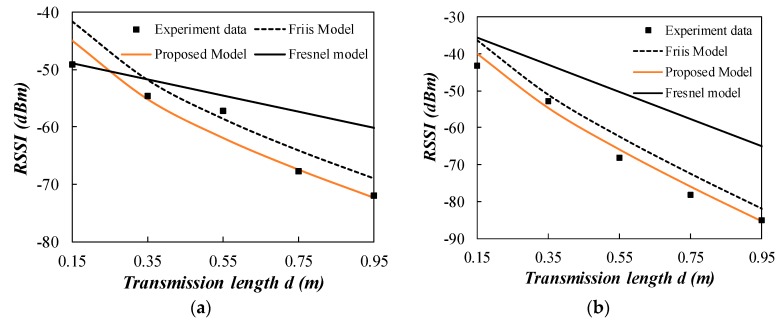
Comparison of the measured experiment data from the literature [32] with the propagation models. (**a**) *σ* = 37.61 mS/m, *ε_r_* = 19 for 8% wet sand (**b**) *σ* = 123.61 mS/m, *ε_r_* = 30 for 15% wet sand.

**Table 1 sensors-20-02580-t001:** Specifications for wireless sensor nodes.

Node Dimension	Antenna Dimension	Operational Frequency	Transmit Power	Omnidirectional Antenna Gains	Receive Sensitivity
200 mm × 120 mm × 80 mm	0.17 m	433 MHz	18.5 dBm	2 dBi	−100 dBm

**Table 2 sensors-20-02580-t002:** Parameters of the backfilled soil.

Soil Type	*ρ_b_*/g·cm^−3^	*ρ_s_*/g·cm^−3^	VWC	*σ_b_*/S·m^−1^	Particle Distribution		
					Sand Particle	Silt Particle	Clay Particle
Clayey silt	1.366	2.72	48.1%	0.400	2.7%	71.0%	26.3%
Wet Sand	1.340	2.69	4.9%	0.001	53.8%	36.6%	9.6%
Dry Sand	1.340	2.69	0.0%	0	53.8%	36.6%	9.6%

**Table 3 sensors-20-02580-t003:** *R*^2^ and Root Mean Squared Error (RMSE) of different models.

Path Loss Model	Clayey Silt-Group 1		Clayey Silt-Group 2		Wet Sand		Dry Sand	
	*R* ^2^	RMSE	*R* ^2^	RMSE	*R* ^2^	RMSE	*R* ^2^	RMSE
Friis model	0.97	3.45	0.92	5.44	−0.27	7.10	−29.1	2.37
Fresnel model	0.87	7.83	0.95	3.91	0.41	4.85	−120	4.77
Proposed two stage model	0.99	1.55	0.99	1.79	0.93	1.63	0.01	0.44
Coefficient *m*	0.837		0.434		0		0.022	

**Table 4 sensors-20-02580-t004:** Soil characteristics of location B and C.

Location	Classification	GWC	*ε*′ (TDR)	*ε*″ (TDR)	*σ*_DC_ (mS/m)
B	Gravelly SAND	17.02%	10.21	1.42	3.74
C	Clayey Silt	41.72%	27.42	5.93	61.23

**Table 5 sensors-20-02580-t005:** *R*^2^ and RMSE of the propagation models fitting the experiment data from the literatures.

Path Loss Model	Friis Model		Fresnel Model		Proposed Model		
	*R* ^2^	RMSE	*R* ^2^	RMSE	*R* ^2^	RMSE	Coefficient m
[23] Location B	−2.2	18.1	−0.18	10.9	0.97	1.74	0.20
[23] Location C	0.74	5.50	0.55	7.26	0.93	2.93	1
[31] (a)	0.34	6.08	0.69	4.16	0.95	1.69	0.20
[31] (b)	−5.1	8.51	0.67	1.98	0.98	0.52	0.11
[32] (a)	0.76	4.19	0.28	7.22	0.89	2.79	1
[32] (b)	0.90	4.97	−0.07	16.06	0.98	2.22	1

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
