# Peer review of "Theoretical and Experimental Studies on the Signal Propagation in Soil for Wireless Underground Sensor Networks"

_sensors, 2020, doi:10.3390/s20092580_

Round 1

Reviewer 1 Report

This paper considers the problem of the attenuation characteristics of wireless signal transmission in soil. The problem itself seems to be challenging.

The authors review and compare two existing propagation models for predicting signal path loss. Both two models have its pros and cons.

A two stage propagation model in soil is proposed in this paper, which consists of a modified Fresnel model in the near field region and the existing Friis model in the far field region. It can be expressed as a uniform equation by means of a coefficient m.

The proposed propagation model is verified by the field experiments conducted by this paper.

Weakenesses:

The writing of this paper is bad seeing from, for example, the abstract and introduction.

Some important details are not explained clearly such as the problem, the challenges, the key contribution, and the technical depth of this paper,. It is difficult to capture the key contribution of this paper.

Does only three times of measurements of the RSSI guarantee repeatability? In my opinion, taking the average of multiple measurements will be a more convincing result.

There is a serious typographical error in this paper. Figure 7 appears three times in Section 5.

There exist some typos. For example,

(1) The semicolon on line 22 of the abstract should be changed to a colon.

(2) The comma on line 94 should be deleted.

(3) The sentence “Measurement of antenna…transition exist between…” should be “Measurement of antenna…transition exists between…”

Reviewer 2 Report

This article proposes a new two-stage model for the wireless propagation of signals in soil. The model consists of two parts: the modified Fresnel model in the near field region and the existing Friis model in the far field region. The two models can be expressed as a uniform equation with the coefficient m. In terms of the structure of the article, the authors introduced the model composition and made corresponding experimental verifications. From the content of the article, this method realizes the dissemination of information in the soil. But there are some issues that authors would like to pay attention to:

1) The innovation of this article is not enough, and there is a lack of innovative summary. For example, in the last part of the introduction, the innovations of this article can be exemplified in the form of 1, 2, and 3 points.

2) Is there a typographical error in part 7? Figure 7 (b) repeats three times.

3) In the third part, the author should be able to add a flowchart or frame diagram to show the novelty of the proposed model.

4) The experimental part only introduces the experimental effect of the proposed model, and lacks the comparison experiment with the existing model.

5) The related work of this article lacks the introduction of the state-of-the-art, and the amount of cited literature is not enough. Authors are advised to improve the introduction of the latest relevant researches to help readers understand this work.

Reviewer 3 Report

1. More deep study can be conducted for the proposed two stage models. 2. The advantages of the proposed two stage model need more descriptions compared to other models.

Reviewer 4 Report

The authors proposed a novel hybrid propagation model for wireless underground sensor network to address signal attenuations problems imposed by the scenario.

Strengths: 

* Math formalization

* Illustrations

Weaknesses:

* Limited motivation

It is a good work and interesting applied research.

Some points need to be clarified to improve the description and work reproducibility:

* There are some typos and blank space missing throughout all paper, e.g. "Frris" (line 356);

* The authors could motivate the importance of the current work with IoT solutions and Smart Farming;

* Some sentences, e.g. "In this paper, the rest of the paper", need to be rewritten, a native English speaker review it is required.

* The main objective sounds like the proposal of the joint method, but it needs to be explicit and well-grounded.

* In the Introduction and in several sentences the objective needs to be clarified. Further, the challenges arose in the text were not retaken to justify the contribution. 

* The authors need to rise more related work to support their contribution and motivation.

* Several acronyms are well-known but need to be explicit in the text.

* It is necessary to describe all type of soils and why they were chosen. What the difference between them? The choice is enough to simulate all possible scenarios?

* The disadvantages of methods were not presented. Also, a comparison between the proposed and Fresnel method could clarify some findings of the work.

* The conclusion and future works can help to improve the current version of the manuscript. These type of information are strictly related to the disadvantage of the proposed method.

* The authors could share openly the acquired data.

Round 2

Reviewer 2 Report

The author has made grate revisions to my questions. I think this revision has greatly improved the quality of the paper. If there are any flaws, I hope that the readers will improve the grammar and language of the paper better.

Reviewer 4 Report

The current version of the paper presents several improvements related to my concerns.

This manuscript is a resubmission of an earlier submission. The following is a list of the peer review reports and author responses from that submission.

Round 1

Reviewer 1 Report

This paper studies wireless signal transmission for WUSNs. The authors conducted on-site transmission experiments to find the most reasonable propagation model, and they also had some interesting observations.

The weakness of this paper is listed as follows:

The authors only did very little experiments, which cannot support establishing a new signal transmission model. For example, each RSSI measurement is only repeated for three times; and there are only two experiment groups, wet and dry. The findings based on the experiments are very trivial. No interesting results are given, and the proposed new model is a simple mixture of existing ones. I don't understand the point of Section 6 titled "Parametric Study". It seems that no experiments are conducted in this section, and the curves are plotted according to the given model. I suppose the whole section can be condensed into a few sentences for discussion. The writing of this paper needs to be polished. 

Overall, I don't see any interesting findings in this paper. The methodology of this paper is quite straightforward and questionable. The results of this paper offer quite limited value for pushing forward the understanding of wireless signal transmission for WUSNs.

Reviewer 2 Report

The Authors studied the signal propagation of wireless signal in soil for wireless underground sensor networks.

In particular, they focused on comparing existing Friis-Peplinski and Fresnel-CRIM models. Based on these models, they proposed another two models, Friis-CRIM and Fresnel-Modified Peplinski models.

From their experimental results, Fresnel-CRIM model seem to be the most suitable propagation to use for WUSNs. In addition, they also observed that the amount of soluble salts and VWC contribute the most to signal attenuation. This was further verified for dry soil by experiment and model predictions. Lastly, the maximum internodal distance is simulated for different types of soil based on the selected Fresnel-CRIM model.

The research is quite interesting and contributes to the domain of WUSNs. However before eventual publication, a minor revision is required. In particular,

In (1), antenna losses are generally factored in the gain of the antenna. Thus, the parameter L in the modified Friis model cannot include antenna losses. If L includes antenna losses, it cannot be set to 1 as there is no practical antenna with no antenna losses. A related work is Abdorahimi, A.M. Sadeghioon Comparison of radio frequency path loss models in soil for wireless underground sensor networks J. Sens. Actuator Netw., 8 (2019), p. 35. Figure 1 shows the comparison between Peplinski model and CRIM. It is not clear which sub-figures are points (1) to (3) are referring to. In addition, are the graphs in sub-figures (c) and (d) for wet or dry sand? In the experiment, respective orientation of the antennas in each node is not stated. The polarization will affect the Friis model if they are not the same. Could the authors indicate the orientation of the antenna in the nodes? In the experiment, node 1 is placed in the soil while node 2 is placed out of the soil. How does this affect the propagation model and maximum distance if there is a signal transition between soil and air? How is the distribution uniformity of water in the wet sand for these experiments different from those conducted horizontally?

Reviewer 3 Report

The authors have presented a topic well investigated by other researchers. Nevertheless, following suggestions should be considered for better readability and to improve the work.

Authors should provide a table containing all symbols and its description. Please provide detailed explanation for the communication/data transmission steps for the experiments. How many packets have been used to complete a one experiment?  Please provide the details on the sensor nodes and gateway used. What kind of protocol was followed? How you calculated the RSSI ? Is it averaged over all 3 measurements? Authors should also revise the draft and mention about the calculation of Rand RMSE for better readability. There is no explanation on the simulation tool used for the parameter analsysis. How you performed simulation? Which type of polarization was used with the sensors for experiment? It is also advised that authors should restructure the Introdcution's last paragrapgh and make it accordingly to the paper organization. The VWC should be "Volumetric Water Content" There is a typo for RSME, it should be RMSE. A detailed proofread is also suggested. Please also provide a discussion paragrapgh considering your results with the already published work and how this work is advancing the state-of-the-art, as the models considered have been extensively used by the communiyty for the soil propagation and experiments.

Round 2

Reviewer 1 Report

It seems that the authors didn't well address my concerns, so I still hold that the paper should be rejected.

First of all, I believe the main (or even whole) contribution of this paper lies in establishing a new signal transmission model, because the latter simulation is very straightforward illustration of different aspects of this model, and cannot offer insightful results.

Second, I firmly believe that it should be very discreet for proposing a new model, especially when the goal is to overturn existing ones. On one hand, comprehensive and sufficient field experiments should be conducted to support such a model, such as with more different values of sand particle, silt particle, clay particle, and even other potential influential parameters like temperature and humidity. On the other hand, besides of experiments, I believe in-depth analysis and discussions about the disadvantages of existing models and advantages of the newly proposed model should be presented. Why a hybrid model based on the two existing models works?

Third, it seems that the authors directly used the parameters of the backfilled soil listed in Table 1 for the latter analysis, which is critical for the establishment of the new model. But do they really reflect the truth? If not, I suppose the authors must take the error into consideration when establishing the new model. After all, I believe a rigorous way is to use dedicated equipments to accurately measure these values.

Reviewer 3 Report

Please correct the accornym use. Once you define the acronym there is no need to use the full again and again. Rest is satisfactory.